# Hierarchical Adversarially Learned Inference

## Abstract

We propose a novel hierarchical generative model with a simple Markovian structure and a corresponding inference model. Both the generative and inference model are trained using the adversarial learning paradigm. We demonstrate that the hierarchical structure supports the learning of progressively more abstract representations as well as providing semantically meaningful reconstructions with different levels of fidelity. Furthermore, we show that minimizing the Jensen-Shanon divergence between the generative and inference network is enough to minimize the reconstruction error. The resulting semantically meaningful hierarchical latent structure discovery is exemplified on the CelebA dataset. There, we show that the features learned by our model in an unsupervised way outperform the best handcrafted features. Furthermore, the extracted features remain competitive when compared to several recent deep supervised approaches on an attribute prediction task on CelebA. Finally, we leverage the model's inference network to achieve state-of-the-art performance on a semi-supervised variant of the MNIST digit classification task.

## 1 Introduction

Deep generative models represent powerful approaches to modeling highly complex high-dimensional data. There has been a lot of recent research geared towards the advancement of deep generative modeling strategies, including Variational Autoencoders (VAE) (Kingma & Welling, 2013), autoregressive models (Oord et al., 2016a;b) and hybrid models (Gulrajani et al., 2016; Nguyen et al., 2016). However, Generative Adversarial Networks (GANs) (Goodfellow et al., 2014) have emerged as the learning paradigm of choice across a varied range of tasks, especially in computer vision Zhu et al. (2017), simulation and robotics Finn et al. (2016) Shrivastava et al. (2016). GANs cast the learning of a generative network in the form of a game between the generative and discriminator networks. While the discriminator is trained to distinguish between the true and generated examples, the generative model is trained to fool the discriminator. Using a discriminator network in GANs avoids the need for an explicit reconstruction-based loss function. This allows this model class to generate visually sharper images than VAEs while simultaneously enjoying faster sampling than autoregressive models.

Recent work, known as either ALI (Dumoulin et al., 2016) or BiGAN (Donahue et al., 2016), has shown that the adversarial learning paradigm can be extended to incorporate the learning of an inference network. While the inference network, or encoder, maps training examples $x$ to a latent space variable $z$, the decoder plays the role of the standard GAN generator mapping from space of the latent variables (that is typically sampled from some factorial distribution) into the data space. In ALI, the discriminator is trained to distinguish between the encoder and the decoder, while the encoder and decoder are trained to conspire together to fool the discriminator. Unlike some approaches that hybridize VAE-style inference with GAN-style generative learning (e.g. Larsen et al. (2015), Chen et al. (2016)), the encoder and decoder in ALI use a purely adversarial approach. One big advantage of adopting an adversarial-only formalism is demonstrated by the high-quality of the generated samples. Additionally, we are given a mechanism to infer the latent code associated with a true data example.

One interesting feature highlighted in the original ALI work (Dumoulin et al., 2016) is that even though the encoder and decoder models are never explicitly trained to perform reconstruction, this can nevertheless be easily done by projecting data samples via the encoder into the latent space, copying

these values across to the latent variable layer of the decoder and projecting them back to the data space. Doing this yields reconstructions that often preserve some semantic features of the original input data, but are perceptually relatively different from the original samples. These observations naturally lead to the question of the source of the discrepancy between the data samples and their ALI reconstructions. Is the discrepancy due to a failure of the adversarial training paradigm, or is it due to the more standard challenge of compressing the information from the data into a rather restrictive latent feature vector? Ulyanov et al. (2017) show that an improvement in reconstructions is achievable when additional terms which explicitly minimize reconstruction error in the data space are added to the training objective. Li et al. (2017b) palliates to the non-identifiability issues pertaining to bidirectional adversarial training by augmenting the generator's loss with an adversarial cycle consistency loss.

In this paper we explore issues surrounding the representation of complex, richly-structured data, such as natural images, in the context of a novel, hierarchical generative model, Hierarchical Adversarially Learned Inference (HALI), which represents a hierarchical extension of ALI. We show that within a purely adversarial training paradigm, and by exploiting the model's hierarchical structure, we can modulate the perceptual fidelity of the reconstructions. We provide theoretical arguments for why HALI's adversarial game should be sufficient to minimize the reconstruction cost and show empirical evidence supporting this perspective. Finally, we evaluate the usefulness of the learned representations on a semi-supervised task on MNIST and an attribution prediction task on the CelebA dataset.

## 2 Related work

Our work fits into the general trend of hybrid approaches to generative modeling that combine aspects of VAEs and GANs. For example, Adversarial Autoencoders (Makhzani et al., 2015) replace the Kullback-Leibler divergence that appears in the training objective for VAEs with an adversarial discriminator that learns to distinguish between samples from the approximate posterior and the prior. A second line of research has been directed towards replacing the reconstruction penalty from the VAE objective with GANs or other kinds of auxiliary losses. Examples of this include Larsen et al. (2015) that combines the GAN generator and the VAE decoder into one network and Lamb et al. (2016) that uses the loss of a pre-trained classifier as an additional reconstruction loss in the VAE objective. Another research direction has been focused on augmenting GANs with inference machinery. One particular approach is given by Dumoulin et al. (2016); Donahue et al. (2016), where, like in our approach, there is a separate inference network that is jointly trained with the usual GAN discriminator and generator. Karaletsos (2016) presents a theoretical framework to jointly train inference networks and generators defined on directed acyclic graphs by leverage multiple discriminators defined nodes and their parents. Another related work is that of Huang et al. (2016b) which takes advantage of the representational information coming from a pre-trained discriminator. Their model decomposes the data generating task into multiple subtasks, where each level outputs an intermediate representation conditioned on the representations from higher level. A stack of discriminators is employed to provide signals for these intermediate representations. The idea of stacking discriminator can be traced back to Denton et al. (2015) which used used a succession of convolutional networks within a Laplacian pyramid framework to progressively increase the resolution of the generated images.

## 3 Hierachical Adversarially Learned Inference

The goal of generative modeling is to capture the data-generating process with a probabilistic model. Most real-world data is highly complex and thus, the exact modeling of the underlying probability density function is usually computationally intractable. Motivated by this fact, GANs (Goodfellow et al., 2014) model the data-generating distribution as a transformation of some fixed distribution over latent variables. In particular, the adversarial loss, through a discriminator network, forces the generator network to produce samples that are close to those of the data-generating distribution. While GANs are flexible and provide good approximations to the true data-generating mechanism, their original formulation does not permit inference on the latent variables. In order to mitigate this, Adversarially Learned Inference (ALI) (Dumoulin et al., 2016) extends the GAN framework to include an inference network that encodes the data into the latent space. The discriminator is then

trained to discriminate between the joint distribution of the data and latent causes coming from the generator and inference network. Thus, the ALI objective encourages a matching of the two joint distributions, which also results in all the marginals and conditional distributions being matched. This enables inference on the latent variables.

We endeavor to improve on ALI in two aspects. First, as reconstructions from ALI only loosely match the input on a perceptual level, we want to achieve better perceptual matching in the reconstructions. Second, we wish to be able to compress the observables, $\boldsymbol{x}$, using a sequence of composed features maps, leading to a distilled hierarchy of stochastic latent representations, denoted by $\boldsymbol{z}_1$ to $\boldsymbol{z}_L$. Note that, as a consequence of the data processing inequality(Cover & Thomas, 2012), latent representations higher up in the hierarchy cannot contain more information than those situated lower in the hierarchy. In information-theoretic terms, the conditional entropy of the observables given a latent variable is non-increasing as we ascend the hierarchy. This loss of information can be seen as responsible for the perceptual discrepancy observed in ALI's reconstructions. Thus, the question we seek to answer becomes: How can we achieve high perceptual fidelity of the data reconstructions while also having a compressed latent space that is strongly coupled with the observables? In this paper, we propose to answer this using a novel model, Hierarchical Adversarially Learned Inference (HALI), that uses a simple hierarchical Markovian inference network that is matched through adversarial training to a similarly constructed generator network. Furthermore, we discuss the hierarchy of reconstructions induced by the HALI's hierarchical inference network and show that the resulting reconstruction errors are implicitly minimized during adversarial training. Also, we leverage HALI's hierarchial inference network to offer a novel approach to semi-supervised learning in generative adversarial models.

### 3.1 A Model for Hierarchical Features

Denote by $\mathcal{P}(S)$ the set of all probability measures on some set $S$. Let $T_{Z|X}$ be a Markov kernel associating to each element $x \in X$ a probability measure $\mathbb{P}_{Z|X=x} \in \mathcal{P}(Z)$. Given two Markov kernels $T_{W|V}$ and $T_{V|U}$, a further Markov kernel can be defined by composing these two and then marginalizing over $V$, i.e. $T_{W|V} \circ T_{V|U} : U \to \mathcal{P}(W)$. Consider a set of random variables $\boldsymbol{x}, \boldsymbol{z}_1, \ldots, \boldsymbol{z}_L$. Using the composition operation, we can construct a hierarchy of Markov kernels or *feature transitions* as

$$T_{\boldsymbol{z}_L|\boldsymbol{x}} = T_{\boldsymbol{z}_L|\boldsymbol{z}_{L-1}} \circ \cdots \circ T_{\boldsymbol{z}_1|\boldsymbol{x}}. \tag{1}$$

A desirable property for these feature transitions is to have some form of inverses. Motivated by this, we define the adjoint feature transition as $T^{\dagger}_{\boldsymbol{z}_l|\boldsymbol{z}_{l-1}} = T_{\boldsymbol{z}_{l-1}|\boldsymbol{z}_l}$. From this, we see that

$$T^{\dagger}_{\boldsymbol{z}_L|\boldsymbol{x}} = T_{\boldsymbol{x}|\boldsymbol{z}_L} = T_{\boldsymbol{z}_1|\boldsymbol{z}_2} \circ \cdots \circ T_{\boldsymbol{z}_{L-1}|\boldsymbol{z}_L}. \tag{2}$$

This can be interpreted as the generative mechanism of the latent variables given the data being the "inverse" of the data-generating mechanism given the latent variables. Let $q(\boldsymbol{x})$ denote the distribution of the data and $p(\boldsymbol{z}_L)$ be the prior on the latent variables. Typically the prior will be a simple distribution, e.g. a standard Gaussian $p(\boldsymbol{z}_L) = \mathcal{N}(\mathbf{0} \mid \boldsymbol{I})$.

The composition of Markov kernels in Eq. 1, mapping data samples $\boldsymbol{x}$ to samples of the latent variables $\boldsymbol{z}_L$ using $\boldsymbol{z}_1, \ldots, \boldsymbol{z}_{L-1}$ constitutes the encoder. Similarly, the composition of kernels in Eq. 2 mapping prior samples of $\boldsymbol{z}_L$ to data samples $\boldsymbol{x}$ through $\boldsymbol{z}_{L-1}, \ldots, \boldsymbol{z}_1$ constitutes the decoder. Thus, the joint distribution of the encoder can be written as

$$q(\boldsymbol{x}, \ldots, \boldsymbol{z}_L) = \prod_{l=2}^{L} q(\boldsymbol{z}_l \mid \boldsymbol{z}_{l-1}) \, q(\boldsymbol{z}_1 \mid \boldsymbol{x}) \, q(\boldsymbol{x}), \tag{3}$$

while the joint distribution of the decoder is given by

$$p(\boldsymbol{x}, \ldots, \boldsymbol{z}_L) = p(\boldsymbol{x} \mid \boldsymbol{z}_1) \prod_{l=2}^{L} p(\boldsymbol{z}_{l-1} \mid \boldsymbol{z}_l) \, p(\boldsymbol{z}_L). \tag{4}$$

---

**Algorithm 1** HALI training procedure.

---

$\theta_g, \theta_d \leftarrow$ initialize network parameters
**repeat**
    **for** $m \in \{1, \ldots, M\}$ **do**
        $\hat{z}_0^{(m)} \sim q(x)$                                                   ▷ Sample from the dataset
        $z_L^{(m)} \sim p(z)$                                                       ▷ Sample from the prior
        **for** $l \in \{1, \ldots, L\}$ **do**
            $\hat{z}_l^m \sim q(z_l \mid \hat{z}_{l-1})$                   ▷ Sample from each level in the encoder's hierarchy
        **end for**
        **for** $l \in \{L \ldots 1\}$ **do**
            $z_{l-1}^{(m)} \sim p(z_{l-1} \mid z_l)$               ▷ Sample from each level in the decoder's hierarchy
        **end for**
        $\rho_q^{(m)} \leftarrow D(\hat{z}_0^{(m)}, \ldots, \hat{z}_l^{(m)})$    ▷ Get discriminator predictions on encoder's distribution
        $\rho_p^{(m)} \leftarrow D(z_0^{(m)}, \ldots, z_l^{(m)})$    ▷ Get discriminator predictions on decoder's distribution
    **end for**
    $\mathcal{L}_d \leftarrow -\frac{1}{M}\sum_{i=1}^{M} \log(\rho_q^{(i)}) - \frac{1}{M}\sum_{i=1}^{M} log(1 - \rho_p^{(i)})$    ▷ Compute discriminator loss
    $\mathcal{L}_g \leftarrow -\frac{1}{M}\sum_{i=1}^{M} \log(1 - \rho_q^{(i)}) - \frac{1}{M}\sum_{i=1}^{M} \log(\rho_p^{(i)})$    ▷ Compute generator loss
    $\theta_d \leftarrow \theta_d - \nabla_{\theta_d}\mathcal{L}_d$             ▷ Gradient update on discriminator network
    $\theta_g \leftarrow \theta_g - \nabla_{\theta_g}\mathcal{L}_g$             ▷ Gradient update on generator networks
**until** convergence

---

The encoder and decoder distributions can be visualized graphically as

$$x \underset{T_{x|z_1}}{\overset{T_{z_1|x}}{\rightleftharpoons}} z_1 \underset{T_{z_1|z_2}}{\overset{T_{z_2|z_1}}{\rightleftharpoons}} z_2 \underset{T_{z_2|z_3}}{\overset{T_{z_3|z_2}}{\rightleftharpoons}} \cdots \underset{T_{z_{L-1}|z_L}}{\overset{T_{z_L|z_{L-1}}}{\rightleftharpoons}} z_L$$

Having constructed the joint distributions of the encoder and decoder, we can now match these distributions through adversarial training. It can be shown that, under an ideal (non-parametric) discriminator, this is equivalent to minimizing the Jensen-Shanon divergence between the joint Eq. 3 and Eq. 4, see (Dumoulin et al., 2016). Algorithm 1 details the training procedure.

### 3.2 A HIERARCHY OF RECONSTRUCTIONS

The Markovian character of both the encoder and decoder implies a hierarchy of reconstructions in the decoder. In particular, for a given observation $x \sim p(x)$, the model yields $L$ different reconstructions $\hat{x}_l \sim T_{x|z_l} \circ T_{z_l|x}$ for $l \in \{1, \ldots, L\}$ with $\hat{x}_l$ the reconstruction of the $x$ at the $l$-th level of the hierarchy. Here, we can think of $T_{z_l|x}$ as projecting $x$ to the $l$-th intermediate representation and $T_{x|z_l}$ as projecting it back to the input space. Then, the *reconstruction error* for a given input $x$ at the $l$-th hierarchical level is given by

$$\mathcal{L}^l(x) = \mathbb{E}_{z_l \sim T_{z_l|x}}[-\log(p(x \mid z_l))]. \tag{5}$$

Contrary to models that try to merge autoencoders and adversarial models, e.g. Rosca et al. (2017); Larsen et al. (2015), HALI does not require any additional terms in its loss function in order to minimize the above reconstruction error. Indeed, the reconstruction errors at the different levels of HALI are minimized down to the amount of information about $x$ that a given level of the hierarchy is able to encode as training proceeds. Furthermore, under an optimal discriminator, training in HALI minimizes the Jensen-Shanon divergence between $q(x, z_1, \ldots, z_L)$ and $p(x, z_1, \ldots, z_L)$ as formalized in Proposition 1 below. Furthermore, the interaction between the reconstruction error and training dynamics is captured in Proposition 1.

**Proposition 1.** *Assuming $q(x, z_l)$ is bounded away for zero for all $l \in \{1, \ldots, L\}$, we have that*

$$\mathbb{E}_{x \sim q(x)}[\mathcal{L}^l(x)] - H(x \mid z_l) \leq K\, D_{JS}(p(x, z_1, \ldots, z_L) \,||\, q(x, z_1, \ldots, z_L)), \tag{6}$$

*where $H(x \mid z_l)$ is computed under the encoder's distribution and $K$ is as defined in Lemma 2 in the appendix.*

On the other hand, proposition 2 below relates the intermediate representations in the hierarchy to the corresponding induced reconstruction error.

**Proposition 2.** *For any given latent variable $z_l$,*

$$\mathbb{E}_{\boldsymbol{x} \sim q_{\boldsymbol{x}}}[\mathbb{E}_{\boldsymbol{z} \sim T_{\boldsymbol{z}_l|\boldsymbol{x}}}[-\log p(\boldsymbol{x} \mid \boldsymbol{z}_l)]] \geq H(\boldsymbol{x} \mid \boldsymbol{z}_l) \tag{7}$$

*i.e. the reconstruction error is an upper bound on $H(\boldsymbol{x} \mid \boldsymbol{z}_l)$.*

In summary, Propositions 1 and 2 establish the dynamics between the hierarchical representation learned by the inference network, the reconstruction errors and the adversarial matching of the joint distributions Eq. 3 and Eq. 4. The proofs on the two propositions above are deferred to the appendix. Having theoretically established the interplay between layer-wise reconstructions and the training mechanics, we now move to the empirical evaluation of HALI.

## 4 EMPIRICAL ANALYSIS: SETUP

We designed our experiments with the objective of addressing the following questions: Is HALI successful in improving the fidelity perceptual reconstructions? Does HALI induces a semantically meaningful representation of the observed data? Are the learned representations useful for downstream classification tasks? All of these questions are considered in turn in the following sections.

We evaluated HALI on four datasets, CIFAR10 (Krizhevsky & Hinton, 2009), SVHN (Netzer et al., 2011), ImageNet 128x128 (Russakovsky et al., 2015) and CelebA (Liu et al., 2015). We used two conditional hierarchies in all experiments with the Markov kernels parametrized by conditional isotropic Gaussians. For SVHN, CIFAR10 and CelebA the resolutions of two level latent variables are $z_1 \in \mathbb{R}^{64 \times 16 \times 16}$ and $z_2 \in \mathbb{R}^{256}$. For ImageNet, the resolutions is $z_1 \in \mathbb{R}^{64 \times 32 \times 32}$ and $z_2 \in \mathbb{R}^{256}$.

For both the encoder and decoder, we use residual blocks(He et al., 2015) with skip connections between the blocks in conjunction with batch normalization(Ioffe & Szegedy, 2015). We use convolution with stride 2 for downsampling in the encoder and bilinear upsampling in the decoder. In the discriminator, we use consecutive stride 1 and stride 2 convolutions and weight normalization (Salimans & Kingma, 2016). To regularize the discriminator, we apply dropout every 3 layers with a probability of retention of 0.2. We also add Gaussian noise with standard deviation of 0.2 at the inputs of the discriminator and the encoder.

## 5 EMPIRICAL ANALYSIS I: RECONSTRUCTIONS

One of the desired objectives of a generative model is to reconstruct the input images from the latent representation. We show that HALI offers improved perceptual reconstructions relative to the (non-hierarchical) ALI model.

### 5.1 QUALITATIVE ANALYSIS

First, we present reconstructions obtained on ImageNet. Reconstructions from SVHN and CIFAR10 can be seen in Fig. 7 in the appendix. Fig. 1 highlights HALI's ability to reconstruct the input samples with high fidelity. We observe that reconstructions from the first level of the hierarchy exhibit local differences in the natural images, while reconstructions from the second level of the hierarchy displays global change. Higher conditional reconstructions are more often than not reconstructed as a different member of the same class. Moreover, we show in Fig. 2 that this increase in reconstruction fidelity does not impact the quality of the generative samples from HALI's decoder.

### 5.2 QUANTITATIVE ANALYSIS

We further investigate the quality of the reconstructions with a quantitative assessment of the preservation of perceptual features in the input sample. For this evaluation task, we use the CelebA dataset where each image comes with a 40 dimensional binary attributes vector. A VGG-16 classifier(Simonyan & Zisserman, 2014) was trained on the CelebA training set to classify the individual attributes. This trained model is then used to classify the attributes of the reconstructions from the

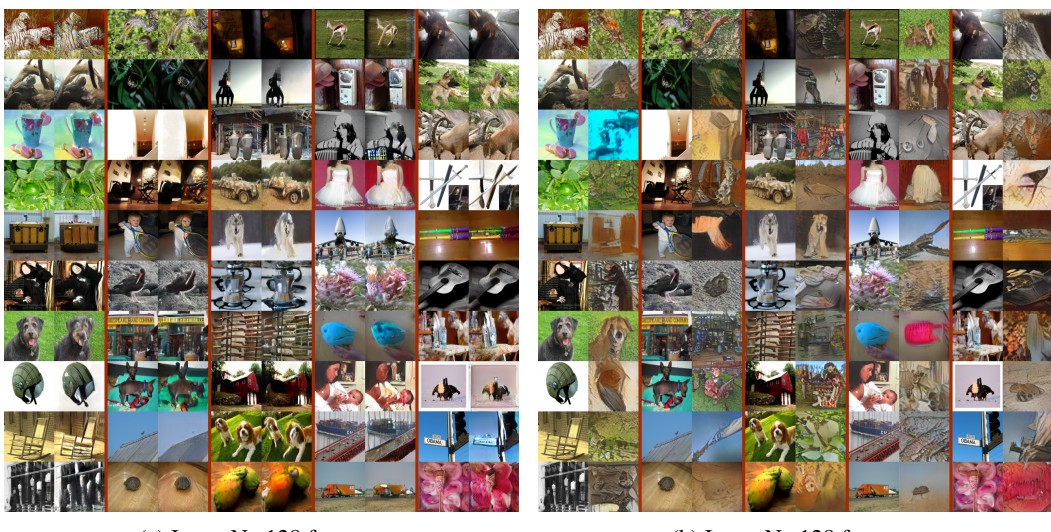

(a) ImageNet128 from $z_1$                    (b) ImageNet128 from $z_2$

Figure 1: ImageNet128 reconstructions from $z_1$ and $z_2$. Odd columns corresponds to examples from the validation set while even columns are the model's reconstructions

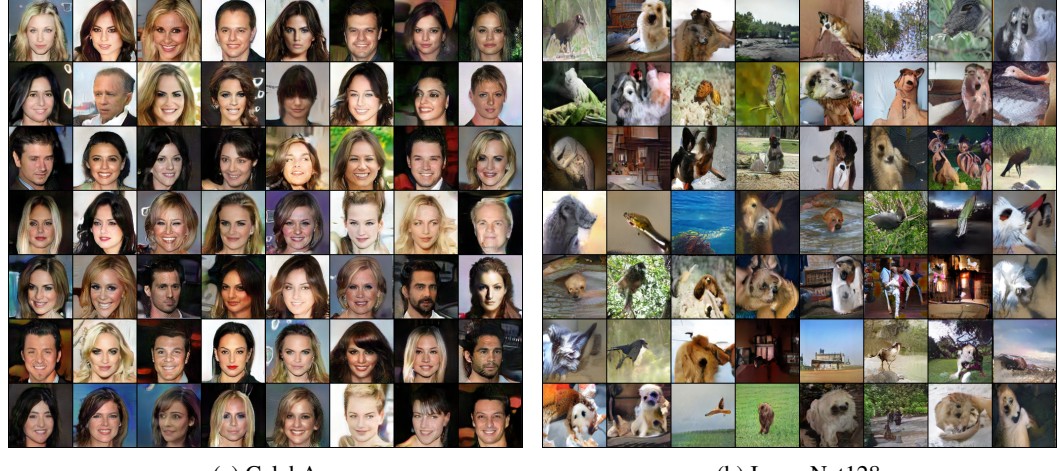

(a) CelebA                    (b) ImageNet128

Figure 2: Samples from $128 \times 128$ CelebA and ImageNet128 datasets

|  | Mean | Std | # Best |
|---|---|---|---|
| Data | 77.13 | 12.48 | |
| VAE | 81.28 | 10.50 | 5 |
| ALI | 84.60 | 5.73 | 3 |
| HALI $z_1$ | 91.35 | 5.62 | 27 |
| HALI $z_2$ | 86.28 | 5.64 | 3 |

Table 1: Summary of CelebA attributes classification from reconstructions for VAE, ALI and the two levels of HALI. The data row is the summary for the VGG classifier and the other scores have been normalized by it. Mean and standard deviation are expressed as percentages. # best represents the count of when a model has the best score on a single attribute. Note that it does not sum to 40 as there were ties.

validation set. We consider a reconstruction as being good if it preserves – as measured by the trained classifier – the attributes possessed by the original sample.

We report a summary of the statistics of the classifier's accuracies in Table 1. We do this for three different models, VAE, ALI and HALI. An inspection of the table reveals that the proportion of attributes where HALI's reconstructions outperforms the other models is clearly dominant. Therefore, the encoder-decoder relationship of HALI better preserves the identifiable attributes compared to other models leveraging such relationships. Please refer to Table 5 in the appendix for the full table of attributes score.

### 5.3 Perceptual Reconstructions

In the same spirit as Larsen et al. (2015), we construct a metric by computing the Euclidean distance between the input images and their various reconstructions in the discriminator's feature space. More precisely, let $\cdot \mapsto \bar{D}(\cdot)$ be the embedding of the input to the pen-ultimate layer of the discriminator. We compute the *discriminator embedded distance*

$$d_c(\boldsymbol{u}, \boldsymbol{v}) = \left\| \bar{D}(\boldsymbol{u}, \hat{\boldsymbol{u}}_1, \hat{\boldsymbol{u}}_2) - \bar{D}(\boldsymbol{v}, \hat{\boldsymbol{v}}_1, \hat{\boldsymbol{v}}_2) \right\|_2, \tag{8}$$

where $\cdot \mapsto \left\| \cdot \right\|_2$ is the Euclidean norm. We then compute the average distances $d_c(\boldsymbol{x}, \hat{\boldsymbol{x}}_1)$ and $d_c(\boldsymbol{x}, \hat{\boldsymbol{x}}_2)$ over the ImageNet validation set. Fig. 3a shows that under $d_c$, the average reconstruction errors for both $\hat{\boldsymbol{x}}_1$ and $\hat{\boldsymbol{x}}_2$ decrease steadily as training advances. Furthermore, the reconstruction error under $d_c$ of the reconstructions from the first level of the hierarchy are uniformly bounded by above by those of the second. We note that while the VAEGAN model of Larsen et al. (2015) explicitly minimizes the perceptual reconstruction error by adding this term to their loss function, HALI implicitly minimizes it during adversarial training, as shown in subsection 3.2.

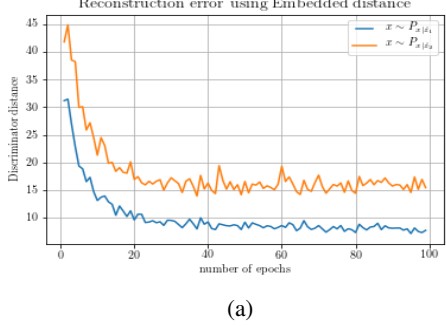
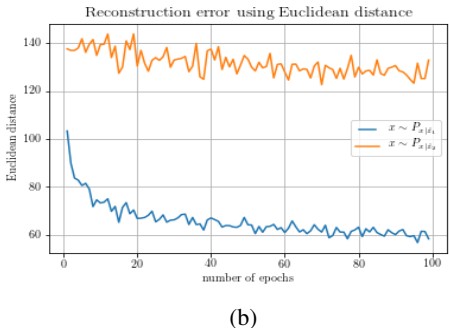

(a)  (b)

Figure 3: Comparison of average reconstruction error over the validation set for each level of reconstructions using the Euclidean (a) and discriminator embedded (b) distances. Using both distances, reconstructions errors for $\boldsymbol{x} \sim T_{\boldsymbol{x}|\boldsymbol{z}_1}$ are uniformly below those for $\boldsymbol{x} \sim T_{\boldsymbol{x}|\boldsymbol{z}_2}$. The reconstruction error using the Euclidean distance eventually stalls showing that the Euclidean metric poorly approximates the manifold of natural images.

## 6 Empirical Analysis II: Learned Representations

We now move on to assessing the quality of our learned representation through inpainting, visualizing the hierarchy and innovation vectors.

## 6.1 INPAINTING

Inpainting is the task of reconstructing the missing or lost parts of an image. It is a challenging task since sufficient prior information is needed to meaningfully replace the missing parts of an image. While it is common to incorporate inpainting-specific training Yeh et al. (2016); Pérez et al. (2003); Pathak et al. (2016), in our case we simply use the standard HALI adversarial loss during training and reconstruct incomplete images during inference time.

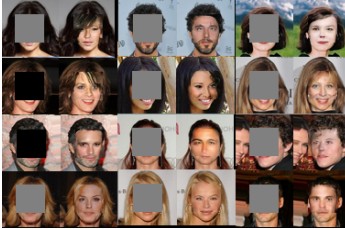 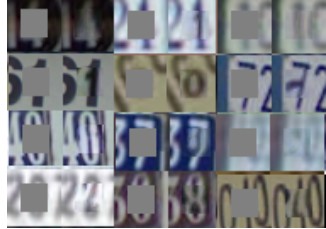 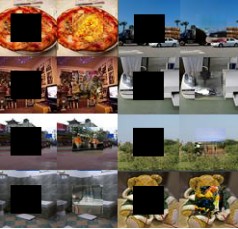

Figure 4: Inpainting on center cropped images on CelebA, SVHN and MS-COCO datasets (left to right).

We first predict the missing portions from the higher level reconstructions followed by iteratively using the lower level reconstructions that are pixel-wise closer to the original image. Fig. 4 shows the inpaintings on center-cropped SVHN, CelebA and MS-COCO (Lin et al., 2014) datasets without any blending post-processing or explicit supervision. The effectiveness of our model at this task is due the hierarchy – we can extract semantically consistent reconstructions from the higher levels of the hierarchy, then leverage pixel-wise reconstructions from the lower levels.

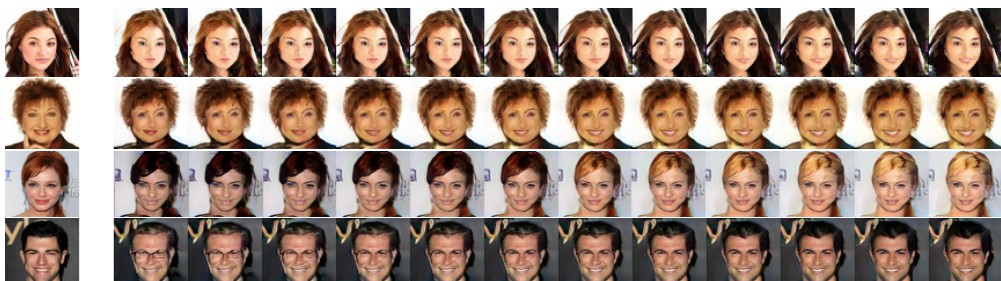

Figure 5: Real CelebA faces (right) and their corresponding innovation tensor (IT) updates (left). For instance, the third row in the figure features Christina Hendricks followed by hair-color IT updates. Similarly, the first two rows depicts usage of smile-IT and the 4th row glasses-plus-hair-color-IT.

## 6.2 HIERARCHICAL LATENT REPRESENTATIONS

To qualitatively show that higher levels of the hierarchy encode increasingly abstract representation of the data, we individually vary the latent variables and observe the effect.

The process is as follow: we sample a latent code from the prior distribution $z_2$. We then multiply individual components of the vector by scalars ranging from $-3$ to 3. For $z_1$, we fix $z_2$ and multiply each feature map independently by scalars ranging from $-3$ to 3. In all cases these modified latent vectors are then decoded back to input data space. Fig. 6 (a) and (b) exhibit some of those decodings for $z_2$, while (c) and (d) do the same for the lower conditional $z_1$. The last column contain the decodings obtained from the originally sampled latent codes. We see that the representations learned in the $z_2$ conditional are responsible for high level variations like gender, while $z_1$ codes imply local/pixel-wise changes such as saturation or lip color.

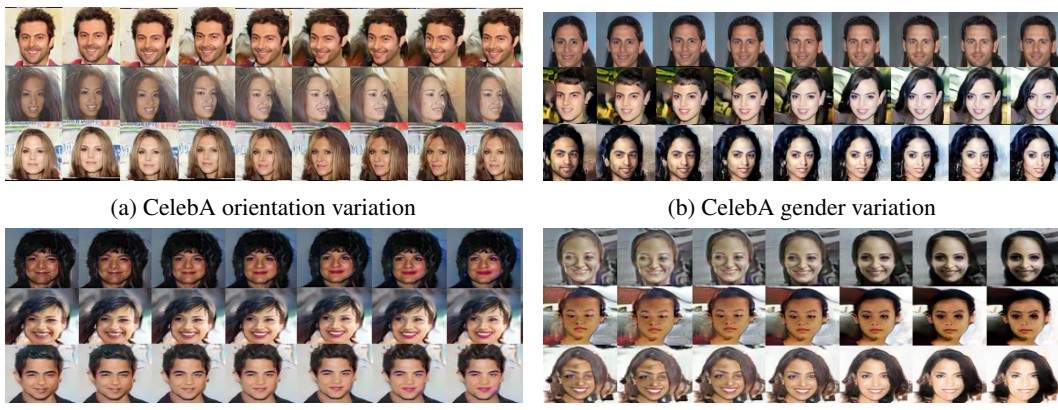

(a) CelebA orientation variation

(b) CelebA gender variation

(c) CelebA lipstick feature map variation

(d) CelebA saturation feature map variation

Figure 6: (a) and (b) showcase $z_2$ vector variation. We sample a set of $z_2$ vectors from the prior. We repeatedly replace a single relevant entry in each vector by a scalar ranging from $-3$ to $3$ and decode. (c) and (d) follows the same process using the $z_1$ latent space.

## 6.3 Latent semantic Innovation

With HALI, we can exploit the jointly learned hierarchical inference mechanism to modify actual data samples by manipulating their latent codes. We refer to these sorts of manipulations as latent semantic innovations.

Consider a given instance from a dataset $x \sim q(x)$. Encoding $x$ yields $\hat{z}_1$ and $\hat{z}_2$. We modify $\hat{z}_2$ by multiplying a specific entry by a scalar $\alpha$. We denote the resulting vector by $\hat{z}_2^\alpha$. We decode the latter and get $\tilde{z}_1^\alpha \sim T_{z_1|z_2}$. We decode the unmodified encoding vector and get $\tilde{z}_1 \sim T_{z_1|\hat{z}_2}$. We then form the *innovation tensor* $\eta^\alpha = \tilde{z}_1 - \tilde{z}_1^\alpha$. Finally, we subtract the innovation vector from the initial encoding, thus getting $\hat{z}_1^\alpha = \hat{z}_1 - \eta^\alpha$, and sample $\tilde{x}^\alpha \sim T_{x|\hat{z}_1^\alpha}$. This method provides explicit control and allows us to carry out these variations on real samples in a completely unsupervised way. The results are shown in Fig. 5. These were done on the CelebA validation set and were not used for training.

## 7 Empirical Evaluations III: Learning Predictive Representations

We evaluate the usefulness of our learned representation for downstream tasks by quantifying the performance of HALI on attribute classification in CelebA and on a semi-supervised variant of the MNIST digit classification task.

### 7.1 Unsupervised classification

Following the protocol established by Berg & Belhumeur (2013); Liu et al. (2015), we train 40 linear SVMs on HALI encoder representations (i.e. we utilize the inference network) on the CelebA validation set and subsequently measure performance on the test set. As in Berg & Belhumeur (2013); Huang et al. (2016a); Kalayeh et al. (2017), we report the *balanced accuracy* in order to evaluate the attribute prediction performance. We emphasize that, for this experiment, the HALI encoder and decoder were trained in on entirely unsupervised data. Attribute labels were only used to train the linear SVM classifiers.

A summary of the results are reported in Table 2. HALI's unsupervised features surpass those of VAE and ALI, but more remarkably, they outperform the best handcrafted features by a wide margin (Zhang et al., 2014). Furthermore, our approach outperforms a number of supervised (Huang et al., 2016a) and deeply supervised (Liu et al., 2015) features. Table 6 in the appendix arrays the results per attribute.

| | Mean | Std | # Best |
|---|---|---|---|
| Triplet-kNN (Schroff et al., 2015) | 71.55 | 12.61 | 0 |
| PANDA (Zhang et al., 2014) | 76.95 | 13.33 | 0 |
| Anet (Liu et al., 2015) | 79.56 | 12.17 | 0 |
| LMLE-kNN (Huang et al., 2016a) | 83.83 | 12.33 | 22 |
| VAE | 73.30 | 9.65 | 0 |
| ALI | 73.88 | 10.16 | 0 |
| HALI | 83.75 | 8.96 | 15 |

Table 2: Summary of statistics for CelebA attributes mean per-class balanced accuracy given in percentage points. # best represents the count of when a model has the best score on a single attribute. Note that it does not sum to 40 as there were ties.

| | MNIST (# errors) |
|---|---|
| VAE (M1+M2) (Kingma et al., 2014) | $233 \pm 14$ |
| VAT (Miyato et al., 2017) | 136 |
| CatGAN (Springenberg, 2015) | $191 \pm 10$ |
| Adversarial Autoencoder (Makhzani et al., 2015) | $190 \pm 10$ |
| PixelGAN (Makhzani & Frey, 2017) | $108 \pm 15$ |
| ADGM (Maaløe et al., 2016) | $96 \pm 2$ |
| Feature-Matching GAN (Salimans et al., 2016) | $93 \pm 6.5$ |
| Triple GAN (Li et al., 2017a) | $91 \pm 58$ |
| GSSLTRABG (Dai et al., 2017) | $79.5 \pm 9.8$ |
| HALI (ours) | **73** |

Table 3: Comparison on semi-supervised learning with state-of-the-art methods on MNIST with 100 labels instance per class. Only methods without data augmentation are included.

## 7.2 SEMI-SUPERVISED LEARNING WITHIN HALI

The HALI hierarchy can also be used in a more integrated semi-supervised setting, where the encoder also receives a training signal from the supervised objective. The currently most successful approach to semi-supervised in adversarially trained generative models are built on the approach introduced by Salimans et al. (2016). This formalism relies on exploiting the discriminator's feature to differentiate between the individual classes present in the labeled data as well as the generated samples. Taking inspiration from (Makhzani et al., 2015; Makhzani & Frey, 2017), we adopt a different approach that leverages the Markovian hierarchical inference network made available by HALI,

$$\boldsymbol{x} \rightarrow \boldsymbol{z} \rightarrow \boldsymbol{y}, \tag{9}$$

Where $\boldsymbol{z} = enc(\boldsymbol{x} + \sigma\,\boldsymbol{\epsilon})$, with $\boldsymbol{\epsilon} \sim \mathcal{N}(0, \boldsymbol{I})$, and $\boldsymbol{y}$ is a categorical random variable. In practice, we characterize the conditional distribution of $\boldsymbol{y}$ given $\boldsymbol{z}$ by a softmax. The cost of the generator is then augmented by a supervised cost. Let us write $\mathcal{D}_{sup}$ as the set of pairs all labeled instance along with their label, the supervised cost reads

$$\mathcal{L}_{sup} = \frac{1}{|\mathcal{D}_{sup}|} \sum_{\substack{(\boldsymbol{y},\boldsymbol{x})\in\mathcal{D}_{sup} \\ \hat{\boldsymbol{y}}\sim q(\boldsymbol{y}|\boldsymbol{x})}} \sum_k \boldsymbol{y}_k \log(\hat{\boldsymbol{y}}_k). \tag{10}$$

We showcased this approach on a semi-supervised variant of MNIST(LeCun et al., 1998) digit classification task with 100 labeled examples evenly distributed across classes.

Table 3 shows that HALI achieves a new state-of-the-art result for this setting. Note that unlike Dai et al. (2017), HALI uses no additional regularization.

## 8 CONCLUSION AND FUTURE WORK

In this paper, we introduced HALI, a novel adversarially trained generative model. HALI learns a hierarchy of latent variables with a simple Markovian structure in both the generator and inference

networks. We have shown both theoretically and empirically the advantages gained by extending the ALI framework to a hierarchy.

While there are many potential applications of HALI, one important future direction of research is to explore ways to render the training process more stable and straightforward. GANs are well-known to be challenging to train and the introduction of a hierarchy of latent variables only adds to this.

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

# A ARCHITECTURE DETAILS

| Operation | Kernel | Strides | Feature maps | BN/WN? | Dropout | Nonlinearity |
|---|---|---|---|---|---|---|
| $G_{z_1}(x) - 3 \times 128 \times 128$ input | | | | | | |
| Convolution | $3 \times 3$ | $1 \times 1$ | 32 | $\times$ | 0.0 | Leaky ReLU |
| Convolution | $3 \times 3$ | $2 \times 2$ | 64 | BN | 0.0 | Leaky ReLU |
| Resnet Block | $3 \times 3$ | $1 \times 1$ | 64 | BN | 0.0 | Leaky ReLU |
| Resnet Block | $3 \times 3$ | $1 \times 1$ | 64 | BN | 0.0 | Leaky ReLU |
| Convolution | $3 \times 3$ | $2 \times 2$ | 128 | BN | 0.0 | Leaky ReLU |
| Resnet Block | $3 \times 3$ | $1 \times 1$ | 128 | BN | 0.0 | Leaky ReLU |
| Convolution | $3 \times 3$ | $1 \times 1$ | 128 | $\times$ | 0.0 | Leaky ReLU |
| Gaussian Layer | | | | | | |
| $G_{z_2}(z_1) - 64 \times 32 \times 32$ input | | | | | | |
| Convolution | $3 \times 3$ | $2 \times 2$ | 256 | BN | 0.0 | Leaky ReLU |
| Convolution | $3 \times 3$ | $2 \times 2$ | 256 | BN | 0.0 | Leaky ReLU |
| Convolution | $3 \times 3$ | $2 \times 2$ | 512 | BN | 0.0 | Leaky ReLU |
| Resnet Block | $3 \times 3$ | $1 \times 1$ | 512 | BN | 0.0 | Leaky ReLU |
| Convolution | $4 \times 4$ | $valid$ | 512 | BN | 0.0 | Leaky ReLU |
| Convolution | $1 \times 1$ | $1 \times 1$ | 512 | $\times$ | 0.0 | Linear |
| $G_{z_1}(z_2) - 128 \times 1 \times 1$ input | | | | | | |
| Convolution | $1 \times 1$ | $1 \times 1$ | $4 * 4 * 256$ | BN | 0.0 | Leaky ReLU |
| Bilinear Upsampling | | | | | | |
| Resnet Block | $3 \times 3$ | $1 \times 1$ | 256 | BN | 0.0 | Leaky ReLU |
| Bilinear Upsampling | | | | | | |
| Convolution | $3 \times 3$ | $1 \times 1$ | 256 | BN | 0.0 | Leaky ReLU |
| Convolution | $3 \times 3$ | $1 \times 1$ | 128 | BN | 0.0 | Leaky ReLU |
| Bilinear Upsampling | | | | | | |
| Convolution | $3 \times 3$ | $1 \times 1$ | 128 | $\times$ | 0.0 | Leaky ReLU |
| Gaussian Layer | | | | | | |
| $G_x(z_1) - 64 \times 32 \times 32$ input | | | | | | |
| Convolution | $1 \times 1$ | $1 \times 1$ | 64 | BN | 0.0 | Leaky ReLU |
| Resnet Block | $3 \times 3$ | $1 \times 1$ | 64 | BN | 0.0 | Leaky ReLU |
| Bilinear Upsampling | | | | | | |
| Resnet Block | $3 \times 3$ | $1 \times 1$ | 64 | BN | 0.0 | Leaky ReLU |
| Bilinear Upsampling | | | | | | |
| Convolution | $3 \times 3$ | $1 \times 1$ | 64 | BN | 0.0 | Leaky ReLU |
| Convolution | $3 \times 3$ | $1 \times 1$ | 32 | BN | 0.0 | Leaky ReLU |
| Convolution | $1 \times 1$ | $1 \times 1$ | 3 | $\times$ | 0.0 | Tanh |
| $D(x) - 3 \times 128 \times 128$ input | | | | | | |
| Convolution | $3 \times 3$ | $1 \times 1$ | 32 | WN | 0.2 | Leaky ReLU |
| Convolution | $3 \times 3$ | $2 \times 2$ | 64 | WN | 0.5 | Leaky ReLU |
| Convolution | $3 \times 3$ | $2 \times 2$ | 64 | WN | 0.5 | Leaky ReLU |
| Convolution | $3 \times 3$ | $1 \times 1$ | 64 | WN | 0.5 | Leaky ReLU |
| $D(x, z_1) - 128 \times 32 \times 32$ input | | | | | | |
| *Concatenate $D(x)$ and $z_1$ along the channel axis* | | | | | | |
| Convolution | $3 \times 3$ | $1 \times 1$ | 128 | WN | 0.2 | Leaky ReLU |
| Convolution | $3 \times 3$ | $2 \times 2$ | 256 | WN | 0.5 | Leaky ReLU |
| Convolution | $3 \times 3$ | $2 \times 2$ | 256 | WN | 0.5 | Leaky ReLU |
| Convolution | $3 \times 3$ | $2 \times 2$ | 512 | WN | 0.5 | Leaky ReLU |
| Convolution | $4 \times 4$ | $valid$ | 512 | WN | 0.2 | Leaky ReLU |
| $D(x, z_1, z_2) - 512 \times 1 \times 1$ input | | | | | | |
| *Concatenate $D(x, z_1)$ and $z_2$ along the channel axis* | | | | | | |
| Convolution | $1 \times 1$ | $1 \times 1$ | 1024 | $\times$ | 0.5 | Leaky ReLU |
| Convolution | $1 \times 1$ | $1 \times 1$ | 1024 | $\times$ | 0.5 | Leaky ReLU |
| Convolution | $1 \times 1$ | $1 \times 1$ | 1 | $\times$ | 0.5 | Sigmoid |

Table 4: Architecture detail for HALI(unsupervised) on the Imagenet 128 and CelebA 128 Datasets.

## B  PROOFS

**Lemma 1.** *Let $f$ be a valid f-divergence generator. Let $p$ and $q$ be joint distributions over a random vector $\boldsymbol{x}$. Let $\boldsymbol{x}_A$ be any strict subset of $\boldsymbol{x}$ and $\boldsymbol{x}_{-A}$ its complement, then*

$$D_f(p(\boldsymbol{x}) \,||\, q(\boldsymbol{x})) \geq D_f(p(\boldsymbol{x}_A) \,||\, q(\boldsymbol{x}_A)) \tag{11}$$

*Proof.* By definition, we have

$$D_f(p(\boldsymbol{x}) \,||\, q(\boldsymbol{x})) = \mathbb{E}_{\boldsymbol{x}\sim q(\boldsymbol{x})}[f(\frac{p(\boldsymbol{x})}{q(\boldsymbol{x})})] = \mathbb{E}_{\boldsymbol{x}_A\sim q(\boldsymbol{x}_A)}\mathbb{E}_{\boldsymbol{x}_{-A}\sim q(\boldsymbol{x}_{-A}|\boldsymbol{x}_A)}[f(\frac{p(\boldsymbol{x}_A)\,p(\boldsymbol{x}_{-A}\mid\boldsymbol{x}_A)}{q(\boldsymbol{x}_A)\,q(\boldsymbol{x}_{-A}\mid\boldsymbol{x}_A)})]$$

Using that $f$ is convex, Jensen's inequality yields

$$D_f(p(\boldsymbol{x}) \,||\, q(\boldsymbol{x})) \geq \mathbb{E}_{\boldsymbol{x}_A\sim q(\boldsymbol{x}_A)}[f(\mathbb{E}_{\boldsymbol{x}_{-A}\sim q(\boldsymbol{x}_{-A}|\boldsymbol{x}_A)}\frac{p(\boldsymbol{x}_A)\,p(\boldsymbol{x}_{-A}\mid\boldsymbol{x}_A)}{q(\boldsymbol{x}_A)\,q(\boldsymbol{x}_{-A}\mid\boldsymbol{x}_A)})]$$

Simplifying the inner expectation on the right hand side, we conclude that

$$D_f(p(\boldsymbol{x}) \,||\, q(\boldsymbol{x})) \geq \mathbb{E}_{\boldsymbol{x}_A\sim q(\boldsymbol{x}_A)}[f(\frac{p(\boldsymbol{x}_A)}{q(\boldsymbol{x}_A)})]$$

$\square$

**Lemma 2** (Kullback-Leibler's upper bound by Jensen-Shannon). *Assume that $p$ and $q$ are two probability distribution absolutely continuous with respect to each other. Moreover, assume that $q$ is bounded away from zero. Then, there exist a positive scalar $K$ such that*

$$D_{KL}(p \,||\, q) \leq K D_{JS}(p \,||\, q). \tag{12}$$

*Proof.* We start by bounding the Kullblack-Leibler divergence by the $\chi^2$-distance. We have

$$D_{KL}(p \,||\, q) \leq \int \log(\frac{p(\boldsymbol{x})}{q(\boldsymbol{x})})\,dx \leq \log(1 + D_{\chi^2}(p \,||\, q)) \leq D_{\chi^2}(p \,||\, q) \tag{13}$$

The first inequality follows by Jensen's inequality. The third inequality follows by the Taylor expansion. Recall that both the $\chi^2$-distance and the Jensen-Shanon divergences are f-divergences with generators given by $f_{\chi^2}(t) = (t-1)^2$ and $f_{JS}(t) = u\log(\frac{2t}{t+1}) + \log(\frac{2t}{t+1})$, respectively. We form the function $t \mapsto h(t) = \frac{f_{\chi^2}(t)}{f_{JS}(t)}$. $h$ is strictly increasing on $[0,\infty)$. Since we are assuming $q$ to be bounded away from zero, we know that there is a constant $c_1$ such that $q(\boldsymbol{x}) > c_1$ for all $\boldsymbol{x}$. Subsequently for all $\boldsymbol{x}$, we have that $\frac{p(x)}{q(x)} \leq c_2 := \max_x \frac{p(x)}{c_1}$. Thus, for all $x$ we have $h(\frac{p(x)}{q(x)}) \leq K := h(c_2)$ and hence $f_{\chi^2}(\frac{p}{q}) \leq K\,f_{JS}(\frac{p(x)}{q(x)})$. Intergrating with respect to $q$, we conclude

$$D_{KL}(p \,||\, q) \leq D_{\chi^2}(p \,||\, q) \leq K\,D_{JS}(p \,||\, q)$$

$\square$

**Proposition 3.** *Assuming $q(\boldsymbol{x}, \boldsymbol{z}_l)$ and $p(\boldsymbol{x}, \boldsymbol{z}_l)$ are positive for any $l \in \{1, \dots, L\}$. We have*

$$\mathbb{E}_{\boldsymbol{x}\sim q(\boldsymbol{x})}[\mathcal{L}^l(\boldsymbol{x})] - H(\boldsymbol{x} \mid \boldsymbol{z}_l) \leq K D_{JS}(p(\boldsymbol{x}, \boldsymbol{z}_1, \dots, \boldsymbol{z}_L) \,||\, q(\boldsymbol{x}, \boldsymbol{z}_1, \dots, \boldsymbol{z}_L))) \tag{14}$$

*Where $H(\boldsymbol{x} \mid \boldsymbol{z}_l)$ is computed under the encoder's distribution $q(\boldsymbol{x}, \boldsymbol{z}_l)$*

*Proof.* By elementary manipulations we have.

$$\mathbb{E}_{\boldsymbol{x}\sim q(\boldsymbol{x})}[\mathcal{L}^l(\boldsymbol{x})] = D_{KL}(p(\boldsymbol{x}, \boldsymbol{z}_l) \,||\, q(\boldsymbol{x}, \boldsymbol{z}_l)) + H(\boldsymbol{x}_l \mid \boldsymbol{z}_l) - D_{KL}(p(\boldsymbol{z}_l) \,||\, q(\boldsymbol{z}_l))$$

Where the conditional entropy $H(\boldsymbol{x}_l \mid \boldsymbol{z}_l)$ is computed $q(\boldsymbol{x}, \boldsymbol{z}_l)$. By the non-negativity of the KL-divergence we obtain

$$\mathbb{E}_{\boldsymbol{x}\sim q(\boldsymbol{x})}[\mathcal{L}^l(\boldsymbol{x})] \leq D_{KL}(p(\boldsymbol{x}, \boldsymbol{z}_l) \,||\, q(\boldsymbol{x}, \boldsymbol{z}_l)) + H(\boldsymbol{x}_l \mid \boldsymbol{z}_l)$$

Using lemma 2, we have

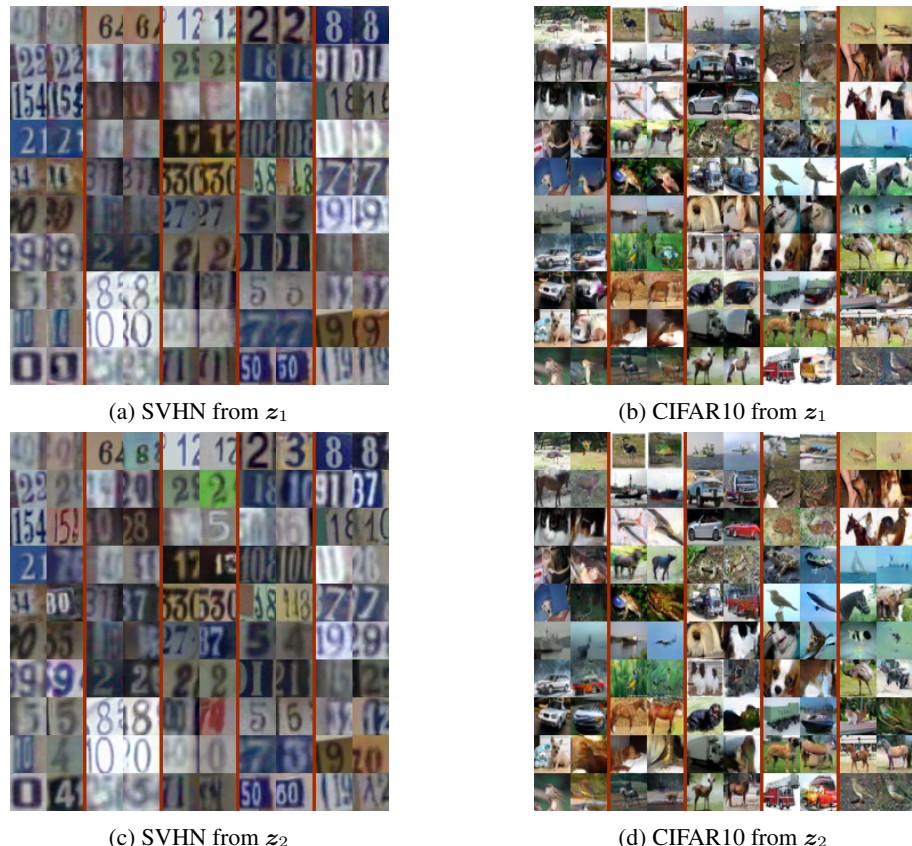

(a) SVHN from $z_1$

(b) CIFAR10 from $z_1$

(c) SVHN from $z_2$

(d) CIFAR10 from $z_2$

Figure 7: Reconstructions for SVHN and CIFAR10 from $z_1$ and reconstructions from $z_2$. Odd columns corresponds to examples from the validation set while even columns are the model's reconstructions

$$\mathbb{E}_{\boldsymbol{x} \sim p_d(\boldsymbol{x})}[\mathcal{L}^l(\boldsymbol{x})] - H(\boldsymbol{x} \mid \boldsymbol{z}_l) \leq KD_{JS}(p(\boldsymbol{x}, \boldsymbol{z}_l) \mid\mid q(\boldsymbol{x}, \boldsymbol{z}_l)))$$

The Jensen-Shanon divergence being f-divergence, using Lemma 1, we conclude

$$\mathbb{E}_{\boldsymbol{x} \sim p_d(\boldsymbol{x})}[\mathcal{L}^l(\boldsymbol{x})] - H(\boldsymbol{x} \mid \boldsymbol{z}_l) \leq KD_{JS}(p(\boldsymbol{x}, \boldsymbol{z}_1, \ldots, \boldsymbol{z}_L) \mid\mid q(\boldsymbol{x}, \boldsymbol{z}_1, \ldots, \boldsymbol{z}_L))).$$

$\square$

**Proposition 4.** *For any given latent variable $\boldsymbol{z}_l$, the reconstruction likelihood* $\mathbb{E}_{\boldsymbol{x} \sim q_{\boldsymbol{x}}}[\mathbb{E}_{\boldsymbol{z} \sim T_{\boldsymbol{z}_l \mid \boldsymbol{x}}}[-\log p(\boldsymbol{x} \mid \boldsymbol{z}_l)]]$ *is an upper bound on $H(\boldsymbol{x} \mid \boldsymbol{z}_l)$.*

*Proof.* By the non-negativity of the Kullback-Leibler divergence, we have that

$$D_{KL}(q(\boldsymbol{x} \mid \boldsymbol{z}_l) \mid\mid p(\boldsymbol{x} \mid \boldsymbol{z}_l)) \geq 0 \implies H[q(\boldsymbol{x} \mid \boldsymbol{z}_l)] \leq \mathbb{E}_{\boldsymbol{z}_l \sim T_{\boldsymbol{z}_l \mid \boldsymbol{x}}}[-\log(p(\boldsymbol{x} \mid \boldsymbol{z}_l))]$$

. Integrating over the marginal and applying Fubini's theorem yields

$$H(\boldsymbol{x} \mid \boldsymbol{z}_l) \leq \mathbb{E}_{\boldsymbol{x} \sim p(x)} \mathbb{E}_{\boldsymbol{z}_l \sim T_{\boldsymbol{z}_l \mid \boldsymbol{x}}}[-\log(p(\boldsymbol{x} \mid \boldsymbol{z}_l))],$$

where the conditional entropy $H(\boldsymbol{x} \mid \boldsymbol{z}_l)$ is computed under the encoder distribution. $\square$

| | Sideburns | Black Hair | Wavy Hair | Young | Makeup | Blond | Attractive | withShadow | withNecktie | Blurry | DoubleChin | BrownHair | Mouth Open | Goatee | Bald | PointyNose | Gray Hair | Pale Skin | ArchedBrows | With Hat |
|---|---|---|---|---|---|---|---|---|---|---|---|---|---|---|---|---|---|---|---|---|
| Data | 86 | 81 | 69 | 91 | 90 | 88 | 81 | 80 | 80 | 60 | 70 | 74 | 92 | 88 | 77 | 60 | 84 | 72 | 69 | 86 |
| VAE | 80 | **96** | 60 | **98** | 79 | 74 | 83 | 74 | **93** | **96** | 73 | 77 | 83 | 77 | 89 | 86 | 69 | 86 | 80 | 84 |
| ALI | 72 | 90 | 83 | 94 | 88 | 81 | 91 | 77 | 83 | 83 | **89** | 76 | 77 | 72 | 83 | 87 | 84 | 79 | 92 | 92 |
| HALI (z1) | **92** | 93 | **93** | 98 | **91** | **95** | **95** | **92** | 89 | 92 | 81 | **95** | **92** | **89** | **94** | **98** | 88 | 90 | **95** | **99** |
| HALI (z2) | 80 | 78 | 91 | 96 | 89 | 94 | 92 | 78 | 90 | 83 | 88 | 80 | 83 | 78 | 88 | 89 | 87 | **91** | 87 | 84 |

| | Balding | StraightHair | Big Nose | Rosy Cheeks | Oval Face | Bangs | Male | Mustache | HighCheeks | No Beard | Eyeglasses | BaggyEyes | WithNecklace | WithLipstick | Big Lips | NarrowEyes | Chubby | Smiling | BushyBrows | WithEarrings |
|---|---|---|---|---|---|---|---|---|---|---|---|---|---|---|---|---|---|---|---|---|
| Data | 72 | 60 | 69 | 71 | 56 | 89 | 97 | 80 | 85 | 96 | 94 | 69 | 54 | 94 | 54 | 58 | 71 | 92 | 72 | 74 |
| VAE | 89 | **94** | 69 | 67 | 83 | 78 | 93 | 82 | 91 | 98 | 73 | 65 | 83 | 92 | 86 | 82 | 78 | 73 | 95 | 64 |
| ALI | 86 | 87 | 89 | **82** | 89 | 76 | 90 | 77 | 86 | 92 | 81 | 87 | 93 | 88 | 83 | 85 | **90** | 85 | 82 | 83 |
| HALI (z1) | **96** | 91 | **90** | 71 | **91** | **94** | **97** | **84** | **96** | 98 | **90** | **94** | 93 | **94** | 83 | **88** | 82 | **97** | **91** | 83 |
| HALI (z2) | 88 | 88 | 89 | 75 | 90 | 84 | 90 | 80 | 89 | 95 | 75 | 86 | **99** | 90 | **88** | 87 | 87 | 89 | 85 | 81 |

Table 5: CelebA attributes accuracies of reconstructions by different models. The data row displays the raw average of positive accuracies, predicting true 1, and negative accuracies, predicting a true 0 by our VGG classifier. Other rows show the same average accuracies where each individual accuracy is normalized by its corresponding data score. The numbers are all percentages.

| | 5 o Clock Shadow | Arched Eyebrows | Attractive | Bags Under Eyes | Bald | Bangs | Big Lips | Big Nose | Black Hair | Blond Hair | Blurry | Brown Hair | Bushy Eyebrows | Chubby | Double Chin | Eyeglasses | Goatee | Gray Hair | Heavy Makeup | High Cheekbones |
|---|---|---|---|---|---|---|---|---|---|---|---|---|---|---|---|---|---|---|---|---|
| Triplet-kNN | 66 | 73 | 83 | 63 | 75 | 81 | 55 | 68 | 82 | 81 | 43 | 76 | 68 | 64 | 60 | 82 | 73 | 72 | 88 | 86 |
| PANDA | 76 | 77 | 85 | 67 | 74 | 92 | 56 | 72 | 84 | 91 | 50 | 85 | 74 | 65 | 64 | 88 | 84 | 79 | 95 | 89 |
| Anet | 81 | 76 | 87 | 70 | 73 | 90 | 57 | 78 | 90 | 90 | 56 | 83 | 82 | 65 | 64 | 95 | 86 | 85 | 96 | 89 |
| LMLE-kNN | 82 | **79** | **88** | 73 | 90 | **98** | 60 | **80** | 92 | **99** | 59 | **87** | 82 | 79 | 74 | **98** | 95 | 91 | **98** | **92** |
| VAE | 78 | 65 | 62 | 68 | 87 | 86 | 58 | 67 | 75 | 83 | 64 | 62 | 72 | 77 | 80 | 81 | 80 | 88 | 75 | 75 |
| ALI | 78 | 70 | 69 | 68 | 89 | 87 | 57 | 69 | 75 | 88 | 65 | 64 | 71 | 78 | 78 | 85 | 79 | 89 | 79 | 64 |
| HALI(Unsup) | **86** | 77 | 80 | **78** | **94** | 93 | **62** | 74 | 85 | 92 | 78 | 77 | 82 | **85** | **86** | 96 | 92 | **93** | 89 | 85 |

| | Male | Mouth Slightly Open | Mustache | Narrow Eyes | No Beard | Oval Face | Pale Skin | Pointy Nose | Receding Hairline | Rosy Cheeks | Sideburns | Smiling | Straight Hair | Wavy Hair | Wearing Earrings | Wearing Hat | Wearing Lipstick | Wearing Necklace | Wearing Necktie | Young |
|---|---|---|---|---|---|---|---|---|---|---|---|---|---|---|---|---|---|---|---|---|
| Triplet-kNN | 91 | 92 | 57 | 47 | 82 | 61 | 63 | 61 | 60 | 64 | 71 | 92 | 63 | 77 | 69 | 84 | 91 | 50 | 73 | 75 |
| PANDA | 99 | 93 | 63 | 51 | 87 | 66 | 69 | 67 | 67 | 68 | 81 | 98 | 66 | 78 | 77 | 90 | 97 | 51 | 85 | 78 |
| Anet | 99 | 96 | 61 | 57 | 93 | 67 | 77 | 69 | 70 | 76 | 79 | 97 | 69 | 81 | 83 | 90 | 95 | 59 | 79 | 84 |
| LMLE-kNN | **99** | **96** | 73 | 59 | **96** | 68 | 68 | 72 | 76 | 78 | 88 | **99** | 73 | 83 | 83 | **99** | **99** | 59 | **90** | **87** |
| VAE | 78 | 67 | 81 | 60 | 79 | 51 | 86 | 59 | 79 | 79 | 79 | 81 | 55 | 69 | 65 | 84 | 78 | 67 | 83 | 69 |
| ALI | 83 | 52 | 82 | 62 | 79 | 54 | 85 | 61 | 78 | 80 | 77 | 60 | 72 | 77 | 67 | 91 | 82 | 67 | 82 | 71 |
| HALI(Unsup) | 96 | 88 | **90** | **72** | 90 | 65 | **89** | 69 | **84** | **89** | **91** | 91 | 70 | 77 | 78 | 95 | 92 | **71** | 89 | 80 |

Table 6: Mean per-class balanced accuracy in percentage points of each of the 40 face attributes on CelebA.

