# OpenReview forum: "Hierarchical Adversarially Learned Inference"
_ICLR.cc/2018/Conference — Reject_

### Official Review · AnonReviewer1 · 2017-11-27
**An interesting idea, but important details are missing**

**Rating:** 5
**Confidence:** 5

**Review:**

_________________________________________________________________________________________________________

I raise my rating on the condition that the authors will also address the minor concerns in the final version, please see details below.
_________________________________________________________________________________________________________

This paper proposes to perform Adversarially Learned Inference (ALI) in a layer-wise manner. The idea is interesting, and the authors did a good job to describe high-level idea, and demonstrate one advantage of hierarchy: providing different levels reconstructions. However, the advantage of better reconstruction could be better demonstrated.  Some major concerns should be clarified before publishing:

(1) How did the authors implement p(x|z) and q(z|x), or p(z_l | z_{l+1}) and q(z_{l+1} | z_l )? Please provide the details, as this is key to the reconstruction issues of ALI.

(2) Could the authors provide the pseudocode procedure of the proposed algorithm? In the current form of the writing, it is not clear what the HALI procedure is, whether (1) one discriminator is used to distinguish the concatenation of (x, z_1, ..., z_L), or (2) L discriminators are used to distinguish the concatenation of (z_l, z_{l+1}) at each layer, respectively?

The above two points are important. If not correctly constructed, it might reveal potential flaws of the proposed technique.

Since one of the major claims for HALI is to provide better reconstruction with higher fidelity than ALI. Could the authors provide quantitative results on MNIST and CIFAR to demonstrate this? The reconstruction issues have first been highlighted and theoretically analyzed in ALICE [*], and some remedy has been proposed to alleviate the issue.  Quantitative comparison on MNIST and CIFAR are also conducted. Could the authors report numbers to compare with them (ALI and ALICE)?

The 3rd paragraph in Introduction should be adjusted to correctly clarify details of algorithms, and reflect up-to-date literature. "One interesting feature highlighted in the original ALI work (Dumoulin et al., 2016) is that ... never explicitly trained to perform reconstruction, this can nevertheless be easily done...". Note that ALI can only perform reconstruction when the deterministic mapping is used, while ALI itself adopted the stochastic mapping. Further, the deterministic mapping is the major difference of BiGAN from ALI. Therefore, more rigorous way to phrase is that "the original ALI work with deterministic mappings", or "BiGAN" never explicitly trained to perform reconstruction, this can nevertheless be easily done... This tiny difference between deterministic/stochastic mappings makes major difference for the quality of reconstruction, as theoretically analyzed and experimentally compared in ALICE. In ALICE, the authors confirmed further source of poor reconstructions of ALI in practice. It would be better to reflect the non-identifiability issues raised by ALICE in Introduction, rather than hiding it in Future Work as "Although recent work designed to improve the stability of training in ALI does show some promise (Chunyuan Li, 2017), more work is needed on this front."

Also, please fix the typo in reference as:
[*] Chunyuan Li, Hao Liu, Changyou Chen, Yunchen Pu, Liqun Chen, Ricardo Henao and Lawrence Carin. ALICE: Towards understanding adversarial learning for joint distribution matching. In Advances in Neural Information Processing Systems (NIPS), 2017.

---

> ### Author Response · Authors · 2018-01-05
> **Answer to AnonReviewer1**
>
> We thank the reviewer for taking the time spent reviewing our paper.
>
> R: “How did the authors implement $p(x|z)$ and $q(z|x)$, or $p(z_l | z_{l+1})$ and $q(z_{l+1} | z_l )$? Please provide the details, as this is key to the reconstruction issues of ALI.”
> A: We apologize for this oversight and add an architecture section in the appendix.
>
> R: “Could the authors provide the pseudocode procedure of the proposed algorithm? In the current form of the writing, it is not clear what the HALI procedure is, whether (1) one discriminator is used to distinguish the concatenation of $(x, z_1, ..., z_L)$, or L discriminators are used to distinguish the concatenation of $(z_l, z_{l+1})$ at each layer, respectively?”
> A: HALI considers the variables $(x, z_1, ..., z_L)$ jointly. Following the reviewer's suggestion we added a pseudocode procedure to the paper.
>
> R: “Since one of the major claims for HALI is to provide better reconstruction with higher fidelity than ALI. Could the authors provide quantitative results on MNIST and CIFAR to demonstrate this? The reconstruction issues have first been highlighted and theoretically analyzed in ALICE [*], and some remedy has been proposed to alleviate the issue.  Quantitative comparison on MNIST and CIFAR are also conducted. Could the authors report numbers to compare with them (ALI and ALICE)?”
>
> A: In order to quantitatively show that HALI yields better reconstruction than ALI on complex large scale dataset, we leveraged the multimodality of the CelebA dataset by computing the proportion of preserved attributes in the different reconstruction level as detected by a pre-trained classifier. The results are shown in the paper (Table 1). Following the reviewer suggestion, we show below the average euclidean error on reconstruction of the CelebA validation set using ALI, ALICE and HALI. We hope that, in conjunction with Table 1, the results below will offer a meaningful proxy to the difficult task of comparing reconstruction errors across models.
>
> Model                        | l2 error
> --------------------------------------
> VAE                            | 18.91
> ALI                             | 53.68
> ALICE(Adversarial)  |92.56
> ALICE(l2)                   | 32.22
> HALI(z_1)                  | 22.74
> HALI(z_2)                  | 48.77
>
>  R: "In ALICE, the authors confirmed further source of poor reconstructions of ALI in practice. It would be better to reflect the non-identifiability issues raised by ALICE in Introduction, rather than hiding it in Future Work as "Although recent work designed to improve the stability of training in ALI does show some promise (Chunyuan Li, 2017), more work is needed on this front."
>  A: Following the reviewer's comment, we now address (Chunyan Li, 2017) in the introduction instead of the conclusion.

---

> > ### Comment · AnonReviewer1 · 2018-01-12
> > **Major concerns are addressed**
> >
> > Thanks for your updates.
> >
> > I am satisfied with responses on two majors concerns: implementation details of conditionals and pseudocode procedure of the proposed algorithm.
> >
> > However, other minor concerns should be addressed in the final version:
> > (1) Comparison of the reconstruction performance on standard datasets: MNIST and CIFAR, on which the quantitative results are reported in ALICE paper. I understand the authors have reported comparison on CelebA validation dataset (on which the quantitative results are NOT reported in ALICE paper). It seems suspicious not to report results on all of them, because it leaves the impression that the comparison is cherry-picked to benefit the proposed method. It is not necessary to be the best on all of them, just honestly benchmark the numbers to have a fair comparison for future research. One can easily combine HALI and the reconstruction regularization in ALICE to achieve better results.
> >
> > (2) The typo in reference is still NOT fixed.
> >
> > I raise my rating to weak acceptance, on the condition that I trust the author will fix the two minor concerns in the final version.

---

> > > ### Author Response · Authors · 2018-01-13
> > > **Answer to AnonReviewer1**
> > >
> > > We thank the reviewer for the prompt response. Following the reviewer's comment, we have fixed the typo in the ALICE reference. We agree that HALI and ALICE could, in theory, be combined to achieve better results and we are currently comparing HALI and ALICE on MNIST and CIFAR-10. We assure the reviewer that the results will be added to the final version of our paper.

---

> > > > ### Comment · AnonReviewer1 · 2018-01-17
> > > > **Concerns are not addressed**
> > > >
> > > > It is a bit disappointing that the authors claim that the typo in the ALICE reference has been fixed, but it is actually not. The current style in the updated version is
> > > >
> > > > "Changyou Chen Yunchen Pu Liqun Chen Ricardo Henao Chunyuan Li, Hao Liu and Lawrence Carin.
> > > > Alice: Towards understanding adversarial learning for joint distribution matching. In Advances in
> > > > Neural Information Processing Systems (NIPS), 2017."
> > > >
> > > > It is still not the correct form (explicitly suggested in the my initial review).
> > > >
> > > > I doubt have much progress would be made before the final version. Also, see my response for your "Answer to AnonReviewer2 review update", in which I have more serious concerns.

---

> > > > > ### Author Response · Authors · 2018-01-18
> > > > > **Answer to concerns not adressed**
> > > > >
> > > > > We thank the reviewer for his feedback.
> > > > >
> > > > > A:"It is a bit disappointing that the authors claim that the typo in the ALICE reference has been fixed, but it is actually not. The current style in the updated version is"
> > > > > R: we apologize for the confusion. The citation was indeed fixed in the bibliography file but did not propagate to the paper. It is now fixed.
> > > > >
> > > > > A:"see my response for your "Answer to AnonReviewer2 review update", in which I have more serious concerns. "
> > > > > B: We have tried to address the reviewer's more serious concerns in our response to "Answer to AnonReviewer2 review update".

---

### Official Review · AnonReviewer2 · 2017-11-27
**The authors propose a hierarchical GAN variant of ALI, but offer little novelty or insights.**

**Rating:** 5
**Confidence:** 5

**Review:**

******
Please note the adjusted review score after revisions and clarifications of the authors.
The paper was improved significantly but still lacks novelty. For context, multi-layer VAEs also were not published unmodified as follow-up papers since the objective is identical. Also, I would suggest the authors study the modified prior with marginal statistics and other means to understand not just 'that' their model performs better with the extra degree of freedom but also 'how' exactly it does it. The only evaluation is sampling from z1 and z2 for reconstruction which shows that some structure is learned in z2 and the attribute classification task. However, more statistical understanding of the distributions of the extra layers/capacity of the model would be interesting.
******

The authors propose a hierarchical GAN setup, called HALI, where they can learn multiple sets of latent variables.
They utilize this in a deep generative model for image generation and manage to generate good-looking images, faithful reconstructions and good inpainting results.

At the heart of the technique lies the stacking of GANS and the authors claim to be proposing a novel model here.
First, Emily Denton et. al proposed a stacked version of GANs in "Deep Generative Image Models using a Laplacian Pyramid of Adversarial Networks", which goes uncited here and should be discussed as it was the first work stacking GANs, even if it did so with layer-wise pretraining.
Furthermore, the differences to another very similar work to that of the authors (StackGan by Huan et al) are unclear and not well motivated.
And third, the authors fail to cite 'Adversarial Message Passing' by Karaletsos 2016, which has first introduced joint training of generative models with structure by hierarchical GANs and generalizes the theory to a particular form of inference for structured models with GANs in the loop.
This cannot be called concurrent work as it has been around for a year and has been seen and discussed at length in the community, but the authors fail to acknowledge that their basic idea of a joint generative model and inference procedure is subsumed there. In addition, the authors also do not offer any novel technical insights compared to that paper and actually fall short in positioning their paper in the broader context of approximate inference for generative models.

Given these failings, this paper has very little novelty and does not perform accurate attribution of credit to the community.
Also, the authors propose particular one-off models and do not generalize this technique to an inference principle that could be reusable.

As to its merits, the authors manage to get a particularly simple instance of a 'deep gan' working for image generation and show the empirical benefits in terms of image generation tasks.
In addition, they test their method on a semi-supervised task and show good performance, but with a lack of details.

In conclusion, this paper needs to flesh out its contributions on the empirical side and position its exact contributions accordingly and improve the attribution.

---

> ### Author Response · Authors · 2018-01-05
> **Answer to AnonReviewer2**
>
> We thank the reviewer for taking the time spent reviewing our paper.
>
> Before we start addressing the reviewer concerns, we would like to stress that the focus of our paper is on providing an adversarially trained generative model with high fidelity reconstructions, useful latent representations, and unsupervised hierarchically organized content discovery. Moreover, we also point out that our approaches does not rely on stacking GANs.
>
> We now answer the reviewer comments.
>
> R: “First, Emily Denton et. al proposed a stacked version of GANs in "Deep Generative Image Models using a Laplacian Pyramid of Adversarial Networks", which goes uncited here and should be discussed as it was the first work stacking GANs, even if it did so with layer-wise pretraining.”
> A: Although our work does not rely on Laplacian pyramids or stacking GANs as presented in Emily Denton  Al. We agree that Emily Denton et. al is an important paper in the context of adversarially trained generative models and correct this oversight by citing the paper.
>
> R: “Furthermore, the differences to another very similar work to that of the authors (StackGan by Huan et al) are unclear and not well motivated.”
> A: “We respectfully point out that both the objectives, training procedure and focus of HALI are significantly different from those of StackGan. StackGan uses a two stage training procedure with distinct discriminators. HALI training is significantly streamlined as we use only one discriminator and one stage. Moreover, contrary to our work, StackGan does not consider the inference problem nor the quality of the learned representations. Moreover,
> Following the reviewer's suggestion, we update the related works section to better situate our work with respect to StackGan.”
>
> R: “And third, the authors fail to cite 'Adversarial Message Passing' by Karaletsos 2016, which has first introduced joint training of generative models with structure by hierarchical GANs and generalizes the theory to a particular form of inference for structured models with GANs in the loop.
> This cannot be called concurrent work as it has been around for a year and has been seen and discussed at length in the community, but the authors fail to acknowledge that their basic idea of a joint generative model and inference procedure is subsumed there.”
> A: First we thank the reviewer for bringing Karaletsos 2016 to our attention and accordingly update our related works section. While Karaletsos 2016 provides an elegant framework to simultaneously train and provide inference for models defined on directed acyclic graphs,  it does not offer any empirical investigation of the proposed model, nor does it consider reconstructions quality, nor the usefulness of the learned hierarchical representations to downstream tasks.
>
> Karaletsos 2016 and our work are significantly different in scope and focus. HALI does not fit in the framework of Karaletsos 2016.
>
> specifically, Karaletsos 2016 matches joint distribution through the use of local discriminators acting on a given variable and its parents. Consider a two level markovian encoder/decoder architecture. Let x, z1, z2 be the variables produced by this architecture. Karaletsos 2016 would use 2 different discriminators, one for the pair (x, z1) and another for the pair (z1, z2). HALI uses one discriminator taking as input the triplet (x, z1, z2). Please note that as consequence of Jensen's inequality Karaletsos 2016 approach will always offer a looser bound on the true Jensen-Shannon divergence during training.
> Figure 1 in Appendix 5.1 of https://arxiv.org/pdf/1506.05751.pdf clearly shows the difference between the two approaches.
>
> We thank the reviewer for the time spent reviewing our work. We have considered your comments in our revised paper. Given the improved paper and our comments, we hope you reconsider your rating.

---

> ### Author Response · Authors · 2018-01-16
> **Answer to AnonReviewer2 review update**
>
> We thank the reviewer for his answer to our clarifications.
>
> R: "The paper was improved significantly but still lacks novelty. For context, multi-layer VAEs also were not published unmodified as follow-up papers since the objective is identical."
> A:   We kindly to point the reviewer to [1], a published work proposing a hierarchical architecture to the VAE.
> We take the liberty to point out that offering an adversarially trained generative model with faithful reconstruction is of significant interest to the community[2][3][4].
> One of our contribution lies in showing that it is possible to do so without adding additional terms to the loss of ALI.
>
> R: "I would suggest the authors study the modified prior with marginal statistics and other means to understand not just 'that' their model performs better with the extra degree of freedom but also 'how' exactly it does it. [...],  However, more statistical understanding of the distributions of the extra layers/capacity of the model would be interesting."
> A: We used information theoretic constructs to highlight the interplay between information compressions, data processing, and reconstruction errors as we move up the hierarchy. This interplay is formalized in proposition 1 and 2 in the paper.
>
> R: "The only evaluation is sampling from z1 and z2 for reconstruction which shows that some structure is learned in z2 and the attribute classification task."
> A: We take the liberty to point out that our empirical set-up does not rely solely on sampling z1 and z2 for reconstructions and the attribute classification task. We used manifold traversal in z1 and z2 to show that the learned representations of samples encoded local information in z1 and global information z2. We exploited this structure in the vector innovation task to show how structure in z2 can be abstracted and carried down to z1 thus allowing semantically meaningful manipulation of test set images. We leveraged the hierarchy and the local/global information dichotomy in the inference network to perform unsupervised image inpainting. We have quantitatively evaluated HALI's reconstructions on the CelebA dataset using an attribute classifier thus showing the superiority of HALI's reconstruction in retaining attributes of the original image when compared to VAE and ALI. Finally, we leveraged the hierarchical inference network in a Semi-supervised learning task.
>
> [1] Philip Bachman. An Architecture for Deep, Hierarchical Generative Models. In Advances in Neural Information Processing Systems (NIPS), 2016.
> [2] Anders Boesen, Lindbo Larsen, Søren Kaae Sønderby, Hugo Larochelle and Ole Winther. Autoencoding beyond pixels using a learned similarity metric. Proceedings of The 33rd International Conference on Machine Learning (ICML), 2016.
> [3] Mihaela Rosca, Balaji Lakshminarayanan, David Warde-Farley, Shakir Mohamed. Variational Approaches for Auto-Encoding Generative Adversarial Networks. arXiv preprint arXiv:1706.04987, 2017.
> [4] Chunyuan Li, Hao Liu, Changyou Chen, Yunchen Pu, Liqun Chen, Ricardo Henao and Lawrence Carin. ALICE: Towards understanding adversarial learning for joint distribution matching. In Advances in Neural Information Processing Systems (NIPS), 2017.

---

> > ### Comment · AnonReviewer1 · 2018-01-17
> > **Concerns on "faithful" reconstruction "without adding additional terms to the loss of ALI"**
> >
> > The authors claim "One of our contribution lies in showing that it is possible to do so (i.e., faithful reconstruction) without adding additional terms to the loss of ALI." It is very much raising my concerns how reliable the results are.
> >
> > Note that recent papers  [1,2] show the original objective of ALI is problematic to learn meaningful mapping. In [1], the authors show that the training objectives of ALI cannot prevent learning meaningless codes for data -- essentially white noise. "Thus if ALI does indeed work then it must be due to reasons as yet not understood, since the training objective can be low even for meaningless solutions". In [2], similar conclusions are shown both theoretically (the non-identifiable issues) and empirically (500+ runs for each algorithm on the toy dataset). The performance variance of ALI is quite large, the probability it yields good solutions is equal to the the probability it yields bad solutions.
> >
> > If HALI shares the same training objective, how could the the problem be alleviated? Perhaps conditioning introduced by the hierarchy reduces entropy? This must be answered confirmedly. Again, one may cherry-pick good solutions to show in the paper, but it is not fully convincing. Multiple runs should be considered to clearly demonstrate it.
> >
> > Also, I agree with Reviewer2 that the novelty of the submission is limited (the proposed model and results are not surprised). I recommended for weak acceptance just because it is a clear paper.
> >
> >
> > [1] S Arora, A Risteski, Y Zhang, arXiv preprint arXiv:1711.02651. Theoretical limitations of Encoder-Decoder GAN architectures.
> >
> > [2] Chunyuan Li, Hao Liu, Changyou Chen, Yunchen Pu, Liqun Chen, Ricardo Henao and Lawrence Carin. ALICE: Towards understanding adversarial learning for joint distribution matching. In Advances in Neural Information Processing Systems (NIPS), 2017.

---

> > > ### Author Response · Authors · 2018-01-18
> > > **Answering the Concerns on "faithful" reconstruction**
> > >
> > > We thank the reviewer for his feedback. We now move to address his concerns about HALI provided more faithful reconstructions.
> > >
> > > R: "The authors claim "One of our contribution lies in showing that it is possible to do so (i.e., faithful reconstruction) without adding additional terms to the loss of ALI." It is very much raising my concerns how reliable the results are"
> > > A: Please note that we are referring here to reconstructions coming from lower levels of the hierarchy. By the data-processing inequality, the information retained by the latent representation is a non-increasing function of the level in the hierarchy. The increased faithfulness of reconstructions coming from lower levels of the hierarchy is quantitatively evaluated on the CelebA validation set by measuring the number of attributes of the original image that are preserved by the reconstruction. Table 1 in the paper shows that reconstructions from z1 preserve a higher number of attributes that reconstructions from z2. Moreover, following [2], we compute the reconstruction errors of the Imagenet 128 validation set under the discriminator's feature map of reconstructions coming from z1 and z2. Figure 3, clearly shows that reconstruction error from z1 is uniformly bounded above by that from z2 and that both reconstruction errors decrease steadily during training. Figure 3 shows that, under both the discriminator's feature space and Euclidean metrics, reconstruction from z1 are closer to the original input image that reconstructions from z2.
> > >
> > > R: " In [1], the authors show that the training objectives of ALI cannot prevent learning meaningless codes for data -- essentially white noise. "Thus if ALI does indeed work then it must be due to reasons as yet not understood since the training objective can be low even for meaningless solutions"
> > > A: We do not claim that all the codes learned by HALI for a given example are meaningful. Rathe,r we claim that latent representations learned by HALI are useful for downstream tasks. We quantitatively demonstrate this claim with an attribute classification task on CelebA and a semi-supervised learning task on MNIST.
> > >
> > > [1] S Arora, A Risteski, Y Zhang, arXiv preprint arXiv:1711.02651. Theoretical limitations of Encoder-Decoder GAN architectures.
> > >
> > > [2] A. B. L. Larsen, S. K. Sønderby, H. Larochelle, and O. Winther. Autoencoding beyond pixels using a learned similarity metric. International Conference on Machine Learning (ICML), 2016.

---

> > ### Comment · AnonReviewer2 · 2018-01-18
> > **Answer to author comments**
> >
> > Dear authors,
> >
> > Thank you for bringing up reference [1] by Bachman. That paper is a perfect example supporting my argument for why I see a lack of novelty in the presented paper.
> > If you look at my comments in detail, I argue that the HALI objective is effectively unchanged from ALI. In fact, it is basically already largely explained in the original ALI paper (v1 section 2.6: https://arxiv.org/pdf/1606.00704v1.pdf) in a way that is easy to implement and follow.
> > To my understanding HALI is an empirically supported version of that section 2.6 without new insights.
> > As such, it is executed well but adds little novelty.
> >
> > In contrast, reference [1] as well as a spiritually related paper by Kingma et al. (Inverse autoregressive flows) tackle the challenge of inferring deep variational autoencoder-type models by changing the inference structures and the objective appropriately instead of just adding a layer.  Reference [1] does this using the Matryoshka structures, while IAF use skip connections gainfully to simplify signal flow during inference.
> > It is precisely that type of work that adds novelty, since it is not a carbon copy of the procedure introduced in the original VAE paper with an extra layer, but represents a meaningful modification to the inference process in order to overcome challenge in phrasing a hierarchical model.
> >
> > HALI, to the best of my understanding, does not change the objective or the inference primitives in a meaningful way and as such the reference [1] is a perfect contrast to HALI exemplifying my comments regarding lack of novelty ( where novelty is defined as researching needed hierarchical versions of the model).
> >
> > In addition, if the paper is aimed more at understanding joint distribution matching, I would recommend that the authors study other cases in addition to image generation to make a more comprehensive case.

---

> > > ### Author Response · Authors · 2018-01-19
> > > **Answer to reviewer's comments**
> > >
> > > We thank the reviewer for his answer.
> > >
> > > While we agree that HALI's objective is effectively unchanged from ALI, we feel that HALI's novelty lies in illustrating how the hierarchy can be leveraged to:
> > >
> > > * Improve reconstructions in adversarially trained generative models.
> > > * Learn a hierarchy of latent representation with increasing levels of abstraction.
> > > * Perform semantic meaningful manipulation on the original image as shown in novel innovation vector transfer and unsupervised image inpainting.
> > >
> > > We thank the reviewer for his feedback.

---

### Official Review · AnonReviewer3 · 2017-11-28

**Rating:** 7
**Confidence:** 3

**Review:**

The paper incorporated hierarchical representation of complex, reichly-structured data to extend the Adversarially Learned Inference (Dumoulin et al. 2016) to achieve hierarchical generative model. The hierarchical ALI (HALI) learns a hierarchy of latent variables with a simple Markovian structure in both the generator and inference. The work fits into the general trend of hybrid approaches to generative modeling that combine aspects of VAEs and GANs.

The authors showed that within a purely adversarial training paradigm, and by exploiting the model’s hierarchical structure, one can modulate the perceptual fidelity of the reconstructions. We provide theoretical arguments for why HALI’s adversarial game should be sufficient to minimize the reconstruction cost and show empirical evidence supporting this perspective.

The performance of HALI were evaluated on four datasets, CIFAR10, SVHN, ImageNet 128x128 and CelebA. The usefulness of the learned hierarchical representations were demonstrated on a semi-supervised task on MNIST and an attribution prediction task on the CelebA dataset. The authors also noted that the introduction of a hierarchy of latent variables can add to the difficulties in the training.

Summary:
——
In summary, the paper discusses a very interesting topic and presents an elegant approach for modeling complex, richly-structured data using hierarchical representation. The numerical experiments are thorough and HALI is shown to generate better results than ALI. Overall, the paper is well written. However, it would provide significantly more value to a reader if the authors could provide more details and clarify a few points. See comments below for details and other points.

Comments:
——
1.	Could the authors comment on the training time for HALI? How does the training time scale with the levels of the hierarchical structure?

2.	How is the number of hierarchical levels $L$ determined? Can it be learned from the data? Are the results sensitive to the choice of $L$?

3.	It seems that in the experimental results, $L$ is at most 2. Is it because of the data or because of the lack of efficient training procedures for the hierarchical structure?

---

> ### Author Response · Authors · 2018-01-05
> **Answer to AnonReviewer3**
>
> We thank the reviewer for taking the time spent reviewing our paper.
>
> We now answer the reviewer’s comments and questions.
>
> R: “Could the authors comment on the training time for HALI? How does the training time scale with the levels of the hierarchical structure?”
> A: The number of hierarchical levels is determined empirically. We did not explore learning the number of Hierarchical levels from the data. In our experiments, we have noticed that additional levels come with decreased training stability.
>
> R:”How is the number of hierarchical levels $L$ determined? Can it be learned from the data? Are the results sensitive to the choice of $L$? It seems that in the experimental results, $L$ is at most 2. Is it because of the data or because of the lack of efficient training procedures for the hierarchical structure?”
> A: Limiting the number of hierarchical levels to 2 allowed for manageable training. Moreover, as the considered datasets come from computer vision, we tried to show that the first level of the hierarchy encoded local structure while the second encoded global properties of the image.

---

### Author Response · Authors · 2018-01-05
**General remark about the positioning of our work**

Before considering the specific comments of the reviewers, We wish to address the general sense that our work lacks novelty. While it's true that we do not offer a novel learning algorithm, we believe that our hierarchical extension of the ALI/BiGAN framework offers an important contribution that is extremely relevant to the current state of the literature on generative models. There are numerous papers (such as Li et al., 2017 -- "ALICE: Towards Understanding Adversarial Learning for Joint Distribution Matching") and even current ICLR submissions (such as "IVE-GAN: INVARIANT ENCODING GENERATIVE ADVERSARIAL NETWORKS")
whose focus is to modify the ALI objective function to improve image reconstruction. We feel our finding that an unmodified but hierarchical ALI model can dramatically improve over ALI reconstructions is timely and will likely have a real impact on future research into generative models.  Our point is made by *not* proposing a novel learning algorithm. It is our hope that the reviewers will consider the utility of our contribution to the developing conversation that is evolving around this sorts of models.

---

### Decision · Program_Chairs · 2018-01-29
**ICLR 2018 Conference Acceptance Decision**

**Decision:**

Reject

**Comment:**

Pros:
- The paper proposes to use a hierarchical structure to address reconstruction issues with ALI model.
- Obtaining multiple latent representations that individually achieve a different level of reconstructions is interesting.
- Paper is well written and the authors made a reasonable attempt to improve the paper during the rebuttal period.

Cons:
- Reviewers agree that the approach lacks novelty as similar hierarchical approaches have been proposed before.
- The main goal of the paper to achieve better reconstruction in comparison to ALI without changing the latter's objective seems narrow. More analysis is needed to demonstrate that the approach out-performs other approaches that directly tackle this problem in ALI.
- The paper does not provide strong arguments as to why hierarchy works (limited to 2 levels in the empirical analysis presented in the paper).
- Semi-supervised learning as a down-stream task is impressive but limited to MNSIT.